# Effect of Ion Corrosion on 517 Phase Stability

**DOI:** 10.3390/ma13245659

**Published:** 2020-12-11

**Authors:** Guijia Wang, Zhiqi Hu, Jun Chang, Yan Guan, Tingting Zhang, Wanli Bi

**Affiliations:** 1Institute of Materials and Metallurgy, University of Science and Technology Liaoning, Anshan 114031, China; anshanwgj@126.com (G.W.); Anshanhzq@126.com (Z.H.); guanyan@ustl.edu.cn (Y.G.); 2Faculty of Infrastructure Engineering, Dalian University of Technology, Dalian 116024, China; mlchang@dlut.edu.cn (J.C.); tingtingzhang@dlut.edu.cn (T.Z.); 3Research Institute of Keda Fengchi Magnesium Building Materials, Anshan 114031, China

**Keywords:** 517 phase, magnesium oxysulfate cement, ion corrosion, mechanical properties, microstructure

## Abstract

The main hydration product and source of strength of magnesium oxysulfate cement is 5Mg(OH)_2_·MgSO_4_·7H_2_O (known as the 517 phase). Hardened pastes containing 92.38% of the 517 phase were synthesized in this study, and the influence of different types of chloride solutions on the stability and compressive strength of the 517 phase was investigated. X-ray diffraction and the Rietveld method were used to investigate the 517 phase transition in chloride solutions. Ion chromatography and inductively coupled plasma spectrometry were used to analyze the ion concentrations of the chloride solutions. Scanning electron microscopy and mercury injection porosimetry were used to investigate the effect of ion erosion on the microstructure and pore size distribution. The results showed that the crystal structure of 517 phase remained stable upon immersion in chloride solutions (except for the CaCl_2_ solution) up to 28 days, and there was no discernible attenuation in the compressive strength of the hardened pastes. Immersion of the 517 phase in CaCl_2_ solution for 28 days caused Ca^2+^ ions to combine with SO_4_^2−^ groups to generate CaSO_4_·2H_2_O, thereby decomposing the 517 phase. An increase in the concentration of magnesium and sulfate ions in the immersion solutions confirmed the decomposition of the 517 phase. Gel-like Mg(OH)_2_ was observed in the microstructure of the decomposed 517 phase, and the decomposition of the 517 phase increased the porosity of the hardened pastes.

## 1. Introduction

Magnesium oxysulfate (MOS) cement is a gas-hardened magnesium cementitious material prepared from active magnesium oxide and a magnesium sulfate aqueous solution [1,2]. MOS cement has the advantages of a light weight, a short setting time [3], and low alkalinity, while providing thermal insulation [4], energy saving, and environmental protection with a low corrosion rate for steel [5]. MOS cement is widely used in light fireproofing and building materials [6]. Unlike magnesium oxychloride (MOC) cement, MOS cement does not does not contain chloride ions and, therefore, does not significantly corrode steel bar; the corrosion rate of MOS cement is even lower than that of Portland cement in the late stages of hydration, such that MOS cement can be used to prepare reinforced concrete. However, poor water resistance limits offshore engineering applications of MOS cement [1,2,6,7,8].

In recent studies, the effect of 517 phase formation on the mechanical properties of MOS cement has been widely investigated. Studies have shown that the 517 phase is an important hydration product of MOS cement. The main method of forming the 517 phase, which increases the strength of MOS cement, is to add a weak acid or a weak acid salt to MOS cement. For example, adding citric acid, sodium citrate [9], tartaric acid [10], and phosphoric acid [6] to MOS cement can significantly improve the mechanical strength and water resistance by promoting the formation of the 517 phase. Additionally, some studies also focused on the stability of 517 phase. Runcevski et al. [2] studied the thermal stability and decomposition of the 517 phase. The 517 phase was found to be stable from 52 °C to 90 °C, and decomposed at or above 90 °C. Ba et al. [11] found that the 517 phase remained stable after carbonation of MOS cement. Zhao [12] showed that the 517 phase was stable in an alkaline solution with pH ≤11 and began to decompose at pH >12, deteriorating the mechanical properties of MOS cement. Wang et al. [13] tested the phase composition and mechanical properties of a hardened paste of MOS cement under dry–wet cycle conditions and found that the 517 phase was stable.

In summary, numerous studies have been conducted on improving the performance of MOS cement and the stability of the hydration product (the 517 phase) both in China and abroad; however, comparatively few studies have been carried out to investigate how ion erosion affects the stability of the 517 phase. The stability of the 517 phase subjected to erosion ions was investigated in this study to expand the application range of MOS cement, especially in the field of marine engineering. A hardened paste of the pure 517 phase was synthesized and characterized, and the compressive strength of the hardened paste was compared after soaking the hardened paste in different chloride solutions. The transformation of the 517 phase during erosion was investigated using X-ray diffraction (XRD) and the Rietveld method, and ion chromatography (IC) and inductively coupled plasma spectrometry (ICP) were used to determine the ion concentration in the solution before and after soaking. The microstructure and pore diameter distribution of the 517 phase before and after soaking in chloride solutions were determined using SEM and mercury intrusion porosimetry (MIP).

## 2. Experiment

### 2.1. Preparation of Raw Materials and Hardened Paste

Figure 1 for schematic of the preparation of raw materials and hardened paste. The 517 phase was synthesized using the following procedure: 100 g of light magnesium oxide (Sinopharm Chemical Reagent Co., Ltd., Shanghai, China) and 122.3 g of magnesium sulfate heptahydrate (Yingkou Magnesite Chemical Ind Group Co., Ltd., Yingkou, China) were mixed evenly; 0.3 g of citric acid (Gongyi Meiyuan Water Purification Co., Ltd., Gongyi, China) was added to 98.3 mL of distilled water to form a solution; the mixed powder was then added to the solution and mixed for approximately 1 min; the resulting slurry was poured into a mold with dimensions of 20 mm × 20 mm × 20 mm, maintained at a temperature of 20 ± 1 °C and a humidity of 60% ± 5%, and demolded after hardening. The hardened paste of the 517 phase was then immersed in different chloride (NaCl, CaCl_2_, AlCl_3_, FeCl_2_, and FeCl_3_) solutions, controlling the concentration of chloride ions in each solution to be 3%.

### 2.2. Test Methods

#### 2.2.1. Compressive Strength

According to national standard GB175-1999, the compressive strength of hardened pastes of the synthesized 517 phase was measured using a YES-2000 universal testing machine. The loading rate was 600 N/s, and the compressive strength was determined for hardened pastes of the 517 phase that were air cured for 168 h and soaked in chloride solutions for 28 days. After a curing period of 168 h, the hardened paste of the 517 phase was immersed in different chloride solutions for a prescribed period, and the compressive strength of the sample was determined; the softening coefficient (*R*_f_) was used to evaluate the corrosion resistance of the hardened paste of the 517 phase. Equation (1) shows the formula used to calculate *R*_f_; to ensure accuracy, three samples were tested in parallel for each group of data, and the test results were averaged.
(1)$$R_{f}=\frac{R_{w}}{R_{a}}$$
where *R*_a_ and *R*_w_ denote the compressive strength of the hardened paste after 168 h of curing and after soaking in a chloride solution for a prescribed time, respectively.

#### 2.2.2. Phase Composition

The XRD and Fourier-transform infrared (FTIR) analysis methods were previously described [9,10]. The samples were analyzed using a Panaco X’pert Powder X-ray diffractometer (XRD, Panalytical X’pert power, Malvern, UK): λ_Cu_ = 0.15418 nm, tube pressure = 40 kV, tube flow = 40 mA, start angle = 5°, end angle = 85°, step size = 0.02°, and time per step = 5 s. The XRD pattern was introduced into High Score software (3.0.5 version, PANalytical B.V, Amelo, Netherlands) and compared with standard diffraction peaks to determine the composition of the 517 phase before and after immersion in the chloride solutions. The samples were then analyzed using an Agilent Technologies Cary 630 FTIR spectrometer (Agilent Technologies, Santa Clara, CA, USA), where the detected wave numbers ranged from 400 to 4000 cm^−1^. The XRD and FTIR analyses were carried out at room temperature.

The amorphous content of the samples was quantified using an internal standard method over 2θ range of 5–85°; a zinc oxide (ZnO) analytic reagent was mixed into the test samples at 15% by mass before analysis [14]. The Rietveld method, as implemented in the Topas 6.0 software (6.0 version, Bruker, Hamburg, Germany), was used for the quantitative analysis of the mineral phases in the 517 phase samples by fitting the peak areas [14]. The Rietveld method has become a powerful tool for quantitative phase analysis by XRD maps. Taylor first used the Rietveld method in 1991 to quantitatively analyze clay minerals [15]. The amorphous contents of the samples in this study were calculated using Equation (2), where *W*_st_ represents the weight fraction of added ZnO, and *R*_st_ is the Rietveld refined weight fraction of ZnO.
(2)$$ACn=\frac{1-W_{\mathrm{st}}/R_{\mathrm{st}}}{100-W_{\mathrm{st}}}\times 10^{4}\%$$

The ionic compositions of the chloride solutions in which the hardened paste was immersed were measured; a Dionex ICS-3000 ion chromatograph (DIONEX, Diane, Sunnyvale, CA, USA) was used to perform a sulfate ion analysis, and a Shimadzu ICPE 9000 spectrometer (Shimadzu, Kyoto, Japonia) was used to perform a magnesium ion analysis, according to the protocol in [12].

#### 2.2.3. Microscopic Morphology and Pore Structure

After being immersed in different solutions, the hardened paste samples of the 517 phase were cut into small pieces with sizes of 3–5 mm and soaked in alcohol to terminate hydration, followed by drying in an oven at 52 °C. The pore diameter distribution of some samples was tested using a mercury intrusion porosimeter (MIP) Auto Pore IV 9500 (MIP, Quantachrome Autoscan 60, Boynton Beach, FL, USA). The remaining samples were analyzed using a (JSM-5610lv) scanning electron microscope (SEM, ZEISS SIGMA HD, Jena, Germany). The sample preparation, curing temperature, and test method were previously described [2].

## 3. Results

### 3.1. Characterization of Synthesized 517 Phase

Figure 2 presents the XRD test results for the synthesized hardened paste, which are similar to the Runcevski et al. [2] crystal data for the 517 phase. A quantitative Rietveld analysis was carried out using Topas 6.0 software (Figure 3). The sample contained 92.38% of the 517 phase and 7.62% of the amorphous phase. The *R*_wp_ was 8.3%; the good Rietveld fine fitting results indicated a 517 phase with relatively high purity. The microstructure of the hardened paste was observed. The matrix was composed of interlaced needle-like/rod-shaped crystals. The whiskers had a diameter of approximately 200–300 nm and a length of 3–5 μm, corresponding to a length-to-diameter ratio of approximately 10 to 25; the O:Mg:S atomic ratio of 72.01:23.91:4.02 was experimentally obtained by testing 30 different clusters of the synthesized 517 phase. Thus, the crystal was confirmed to be the 517 phase.

### 3.2. Compressive Strength of Hardened Paste of 517 Phase

Figure 4 shows the compressive strength and softening coefficient of the hardened paste of the 517 phase after curing in air and soaking in different chloride solutions. The compressive strength of the paste after air curing was measured as 22.76 MPa. After the paste was soaked in NaCl, AlCl_3_, FeCl_2_, and FeCl_3_ solutions for 28 days, the compressive strength did not decrease significantly and was 22.18 MPa, 21.38 MPa, 22.04 MPa, and 21.36 MPa, respectively, whereas the softening coefficient was 0.965, 0.926, 0.941, and 0.934, respectively. Soaking the paste in a CaCl_2_ solution for 28 days caused the compressive strength to decrease to 12.43 MPa, and the softening coefficient was 0.548. The results presented above show that the chloride solutions, except for the CaCl_2_ solution, had little effect on the compressive strength of the hardened paste.

### 3.3. Transformation of 517 Phase in Chloride Solutions

Figure 5 shows the XRD pattern for the 517 phase after the hardened pastes were soaked for 28 days in different chloride solutions. The composition of the pastes before and after soaking in the chloride solutions, except for the CaCl_2_ solution, corresponded to the 517 phase. CaSO_4_·2H_2_O, Mg(OH)_2_ and 517 phases were detected in the samples that were immersed in the CaCl_2_ solution for 28 days.

The phase composition was further confirmed by using the Rietveld method to analyze the XRD patterns obtained using the internal method. Figure 6 shows the fitting result for the hardened pastes of the 517 phase immersed in the CaCl_2_ solution for 28 days. Table 1 is a comparison of the phase composition of the different samples. The chloride solutions, except for the CaCl_2_ solution, had little effect on the 517 phase, where the 517 phase content remained above 90% after soaking for 28 days in the chloride solutions.

Figure 7 shows the infrared spectrum for the hardened paste of the 517 phase before and after immersion in the CaCl_2_ solution. The 4700–2500 cm^−1^ band corresponds to crystalline water and hydroxide anion vibrational bands. The 3720 and 3640 cm^−1^ peaks are MgO–H asymmetric vibration peaks, and the 3400 cm^−1^ peak is the characteristic crystalline water stretching vibration. The 2500–470 cm^−1^ band corresponds to an HO–H vibration peak, while four other sulfate anion vibration peaks can be identified at 1100 cm^−1^, 979 cm^−1^, 657 cm^−1^, and 426 cm^−1^. After soaking the paste in CaCl_2_ solution, the peak strength of the vibrational bands for MgO–H and crystalline water increased significantly, indicating an increase in the content of hydroxide and crystalline water in the system, which is consistent with the XRD results.

Table 2 lists the concentrations of Mg^2+^ and SO_4_^2−^ in the chloride solutions after immersion of the hardened pastes for 28 days. The Mg^2+^ and SO_4_^2−^ concentrations in the NaCl, AlCl_3_, FeCl_2_, and FeCl_3_ immersion solutions were 3.97 and 0.63 mmol/L, 4.03 and 0.67 mmol/L, 3.87 and 0.66 mmol/L, and 3.93 and 0.64 mmol/L, respectively. The molar ratio of Mg^2+^ to SO_4_^2−^ was approximately 6. After immersing the hardened paste of the 517 phase in the CaCl_2_ solution for 28 days, the Mg^2+^ and SO_4_^2−^ concentrations were 37.43 mmol/L and 32.48 mmol/L, respectively. The molar ratio of Mg^2+^ to SO_4_^2−^ was approximately 1. The 517 phase only has a solubility of 0.34 g/L in solution; therefore, only a small quantity of Mg^2+^ and SO_4_^2−^ are dissolved in the CaCl_2_ solution, as expressed by Equation (3). The 517 phase decomposition products are Mg(OH)_2_, MgSO_4_, and H_2_O; as one of the decomposition products of 517 phase, magnesium sulfate is easily soluble in water, the dissolution can be described by Equation (4), where the molar ratio of Mg^2+^ to SO_4_^2−^ is equal to 1. When immersed in CaCl_2_ solution, the Ca^2+^ ions in solution react with dissolved SO_4_^2−^ to form CaSO_4_·2H_2_O, thereby reducing the SO_4_^2−^ concentration and accelerating the decomposition of the 517 phase, as given by Equation (5).
(3)$$5{\mathrm{Mg}(\mathrm{OH})}_{2}\cdot {\mathrm{MgSO}}_{4}\cdot {7H}_{2}O\to {6\mathrm{Mg}}^{2+}+{\mathrm{SO}}_{4}{}^{2-}+10{\mathrm{OH}}^{-}+7H_{2}O$$
(4)$${\mathrm{MgSO}}_{4}\to {\mathrm{Mg}}^{2+}+{\mathrm{SO}}_{4}{}^{2-}$$
(5)$${\mathrm{Ca}}^{2+}+{\mathrm{SO}}_{4}{}^{2-}+2H_{2}O={\mathrm{CaSO}}_{4}\cdot 2H_{2}O$$

### 3.4. Effect of Chloride Solutions on Microstructure and Pore Size Distribution of 517 Phase

Figure 8a shows a SEM image of the 517 phase before immersion in the chloride solutions. The 517 phase crystals were needle-like/rod-shaped and closely arranged in the matrix. Figure 8b shows an SEM image of the 517 phase after immersion in a NaCl solution for 28 days; there was no discernible change in the microstructure of the 517 phase, whereby the crystals remain needle-like/rod-shaped, and no other crystal formation could be observed. SEM images of the 517 phase after immersion in a CaCl_2_ solution for 28 days are shown in Figure 8c,d; the matrix consisted of the 517 phase, CaSO_4_·2H_2_O, and layered Mg(OH)_2_. The morphology of the 517 phase changed, and the surface of the needle-like/rod-shaped crystals was coated with a gelatinous substance, which may be gel-like Mg(OH)_2_, corresponding to the increase in the amorphous content of the sample, as presented in Table 1. The formation of CaSO_4_·2H_2_O and Mg(OH)_2_ created an expansion stress in the matrix, resulting in expansion cracking of the sample, which was the main reason for the decrease in the compressive strength and the softening coefficient of the hardened slurry of the 517 phase [6,16,17].

Figure 9 shows the cumulative porosity of the 517 phase before and after immersion of the hardened paste in the CaCl_2_ solution for 28 days. The pore size of the sample after immersion was larger than before immersion, which was related to the formation of CaSO_4_·2H_2_O and Mg(OH)_2_, resulting in volume expansion cracking. Figure 10 shows the pore size distribution curve before and after immersing the hardened paste of the 517 phase in the CaCl_2_ solution. The pore size distribution before immersion consisted mainly of small capillary pores (10–100 nm), indicating that the 517 phase has good crystallinity. After immersion in the CaCl_2_ solution, the sample contained noticeably large pores (>100 nm), and the average pore size was also larger than before immersion. It can be seen that, when the 517 phase was immersed in CaCl_2_ solution, the number of pores between 10 nm and 100 nm was larger than before immersion, and the increase was attributed to the generation of poorly crystalline products. This corresponded to the result of the increase in the content of amorphous substance, as presented in Table 1. In addition, the increase in the volume fractions of large pores (>100 nm) had a negative effect on the compressive strength of 517 phase hardened paste. The results for the cumulative porosity and the pore size distribution show that immersing the hardened paste in the CaCl_2_ solution increased the content of large pores and decreased the content of small capillary pores, resulting in an increase in the porosity.

## 4. Discussion

Many studies have shown that the 517 phase content determines the performance of magnesium sulfate cement because the 517 phase is the main contributor to the cement strength [6,7,10,11,12]. The test results presented above showed that the 517 phase is stable in NaCl, AlCl_3_, FeCl_2_, and FeCl_3_ solutions but undergoes a decomposition reaction (Equation (6)) in a CaCl_2_ soaking solution. The structural analysis showed that the decomposition products of the 517 phase are Mg(OH)_2_ and MgSO_4_, which increases the Mg^2+^ and SO_4_^2−^ concentrations in the soaking solution. Table 3 shows the $\Delta_{f}G_{m}^{\Theta }$ for each substance in the system its standard state; the values of $\Delta_{f}G_{m}^{\Theta }$ according to Equations (4) and (5) were calculated as −25.26 KJ/mol and −10.07 KJ/mol, showing that the reaction can proceed spontaneously. According to the second law of thermodynamics, chemical reactions always proceed in the direction of negative $\Delta_{f}G_{m}^{\Theta }$, and a more negative value of $\Delta_{f}G_{m}^{\Theta }$ denotes a higher likelihood of the chemical reaction to proceed [18]; thus, the probability of the chemical reactions occurring can be theoretically ranked from the magnitude and sign of the Gibbs free energy of each chemical reaction [19]. Accordingly, Ca^2+^ preferentially combines with the SO_4_^2−^ dissolved in the 517 phase in solution to form CaSO_4_·2H_2_O, which consumes SO_4_^2−^ and accelerates the decomposition of the 517 phase. The formation of gypsum causes expansion, which leads to the appearance of large pores and cracks inside the sample and loosens the matrix, thus reducing the compressive strength. The crystal packing of the 517 phase has a characteristic laminar structure (see Figure 11), and the main blocks consist of infinite triple chains of MgO_6_ octahedra. Between the laminar structures, the stability of the structure is maintained by some charge equilibrium, and the reaction between the Ca^2+^ and the sulfate destroys the equilibrium, which leads to the decomposition of the 517 phase. Other crystals, characterized by a laminar structure, exhibit a similar tendency, such as mono-sulfur sulfoaluminate, which shows a change in its crystal structure in specific environments [20].
(6)$$5{\mathrm{Mg}(\mathrm{OH})}_{2}\cdot {\mathrm{MgSO}}_{4}\cdot 7H_{2}O\to {\mathrm{Mg}}^{2+}+{\mathrm{SO}}_{4}{}^{2-}+5{\mathrm{Mg}(\mathrm{OH})}_{2}+7H_{2}O$$

## 5. Conclusions

In this study, the effect of different chloride solutions on the compressive strength, phase transition, and microstructure of the 517 phase was investigated, and the following conclusions were obtained:The compressive strength of the 517 phase remained stable in NaCl, AlCl_3_, FeCl_2_, and FeCl_3_ solutions but decreased to 12.43 MPa after soaking in a CaCl_2_ solution for 28 days.Immersing the hardened paste in the CaCl_2_ solution caused the 517 phase to decompose into Mg(OH)_2_, MgSO_4_, and H_2_O. According to the second law of thermodynamics, Ca^2+^ preferentially reacted with SO_4_^2−^ to form gypsum, which broke the charge balance of the 517 phase and destroyed its laminar structure, thereby leading to 517 phase decomposition, as reflected by the decrease in 517 phase content in harden pastes and the obvious increase in concentrations of Mg^2+^ and SO_4_^2−^ in immersing solutions. When immersed in other chloride solutions, the 517 phase was stable and no new substance was formed.The formation of gypsum increased the content of large pores (>100 nm), while the decomposition of the 517 phase was accompanied by the appearance of some amorphous materials with poor crystallization, which enlarged the pore size (10–100 nm), thereby increasing the porosity of the hardened paste.

## Figures and Tables

**Figure 1 materials-13-05659-f001:**
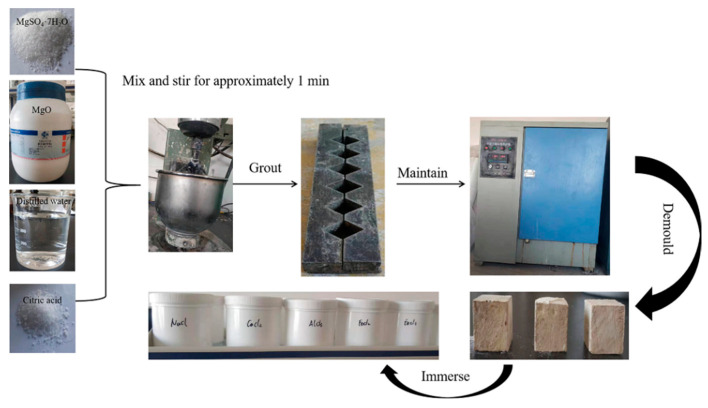
Schematic of the preparation of raw materials and hardened paste.

**Figure 2 materials-13-05659-f002:**
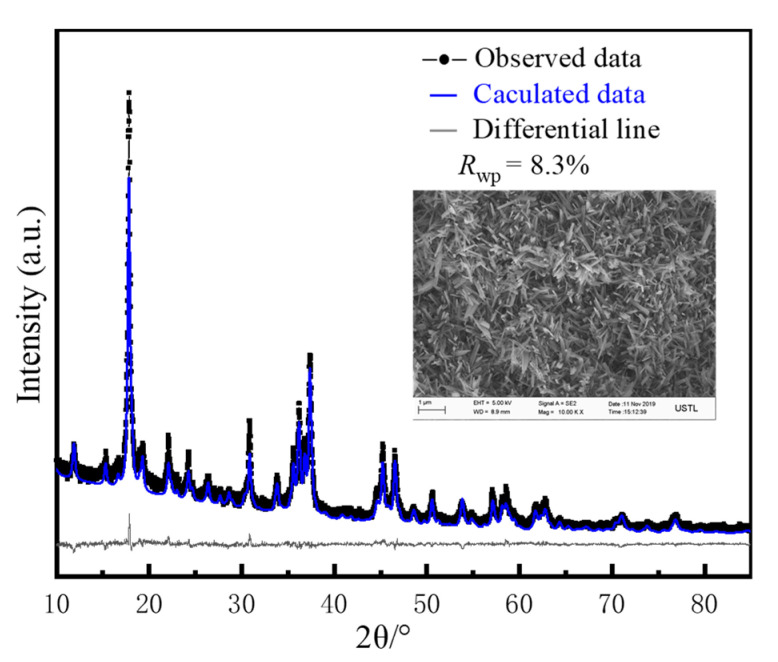
Synthesized hardened slurry: Rietveld diagram obtained by X-ray diffraction (XRD) and microstructure observed by SEM.

**Figure 3 materials-13-05659-f003:**
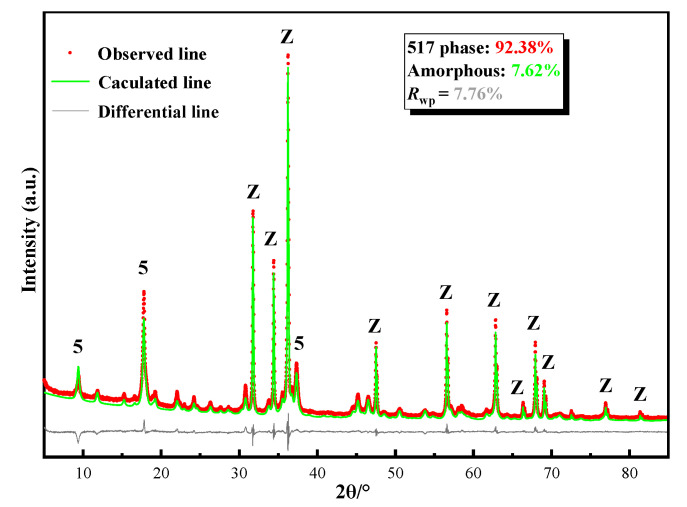
Quantitative Rietveld plots for synthesized hardened paste tested by external standard method: Z:ZnO, 5:517 phase.

**Figure 4 materials-13-05659-f004:**
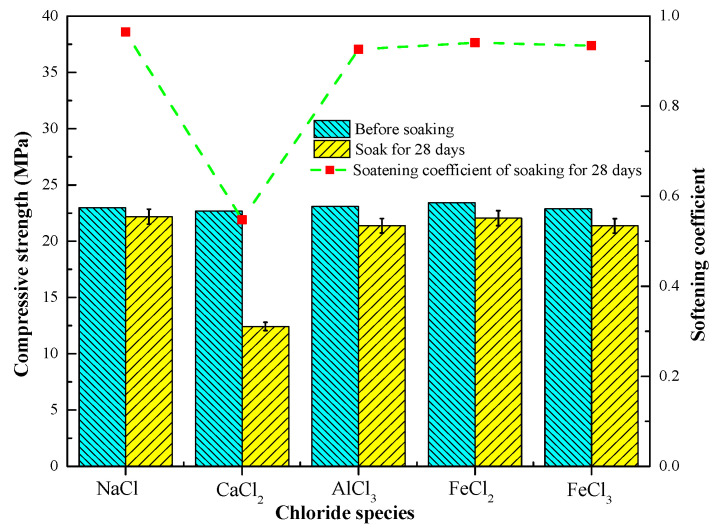
Compressive strength and softening coefficient of hardened pastes of 517 phase immersed in different chloride solutions for 28 days.

**Figure 5 materials-13-05659-f005:**
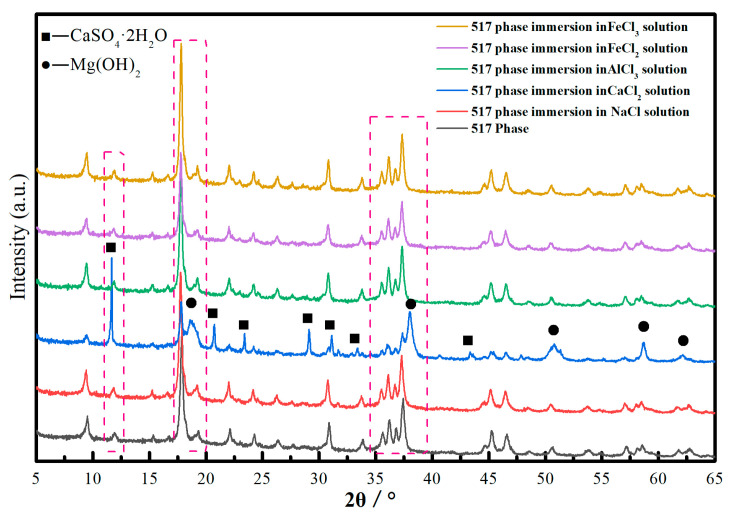
XRD patterns of 517 phase after immersing hardened paste in different chloride solutions for 28 days.

**Figure 6 materials-13-05659-f006:**
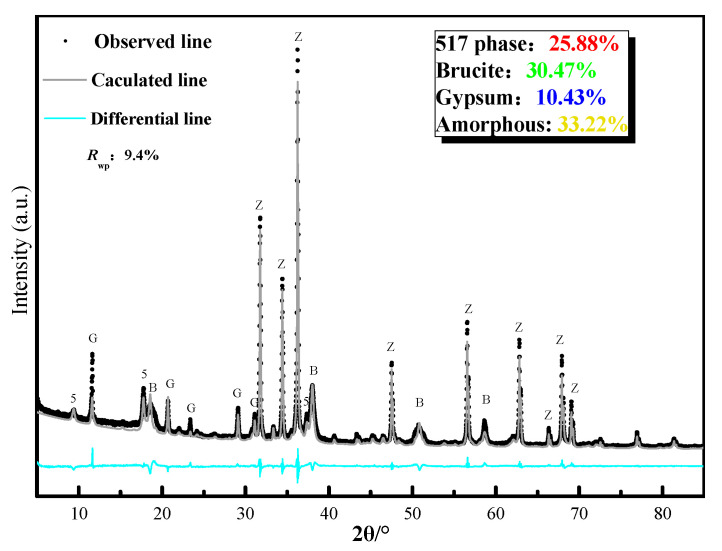
Rietveld fitting results for 517 phase after immersing hardened paste in CaCl_2_ solution for 28 days; Z:ZnO, 5:517 phase, G:CaSO_4_·2H_2_O, B:Mg(OH)_2._

**Figure 7 materials-13-05659-f007:**
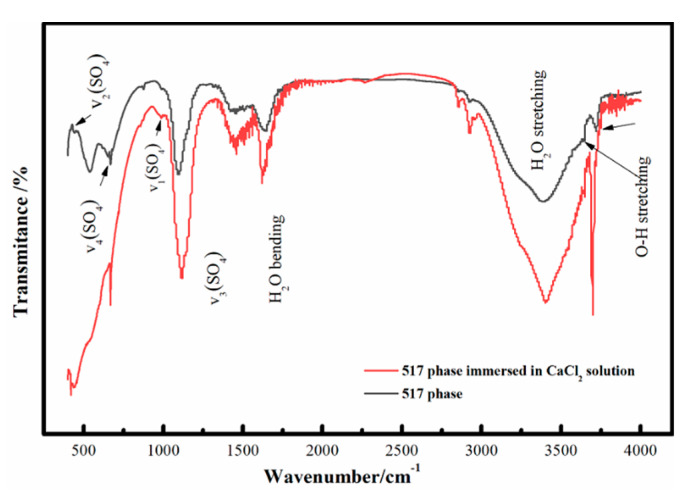
Infrared spectra of 517 phase before and after immersion of hardened paste in CaCl_2_ solution.

**Figure 8 materials-13-05659-f008:**
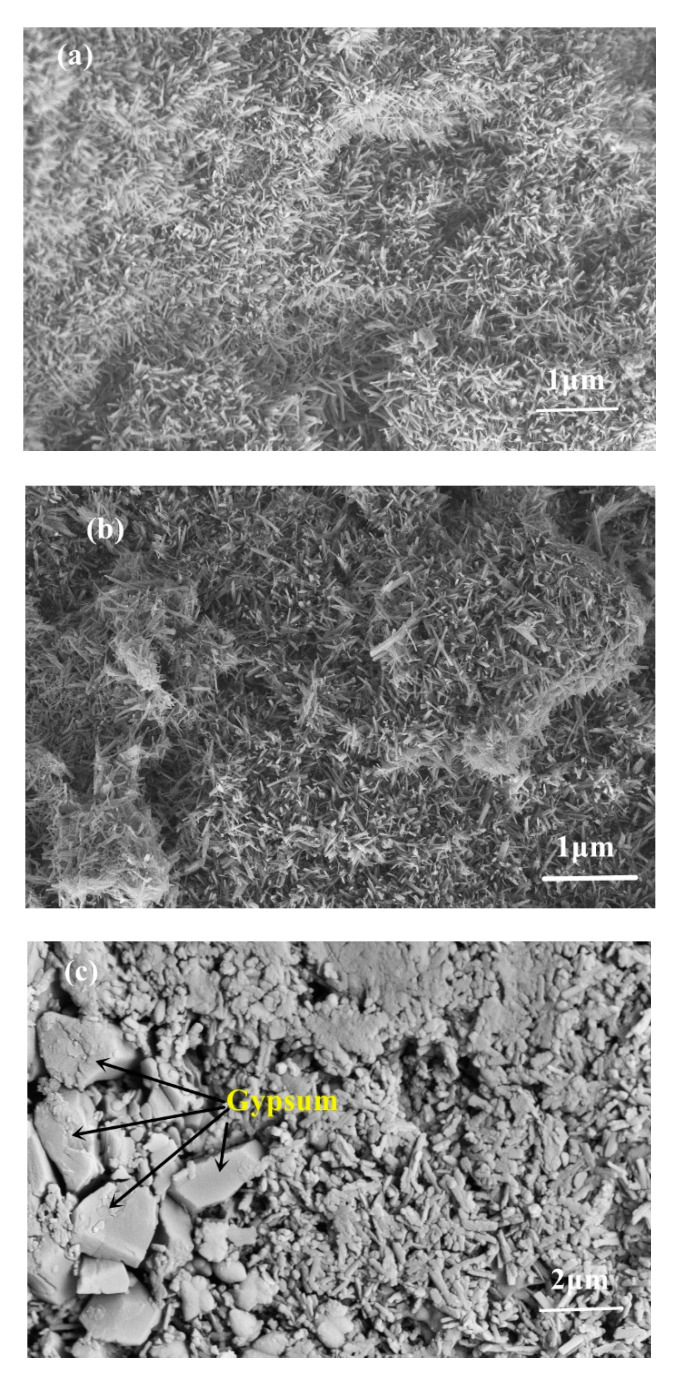
SEM images of 517 phase before and after soaking in different solutions: (**a**) before soaking; (**b**) after soaking in NaCl solution for 28 days; (**c**,**d**) after soaking in CaCl_2_ solution for 28 days.

**Figure 9 materials-13-05659-f009:**
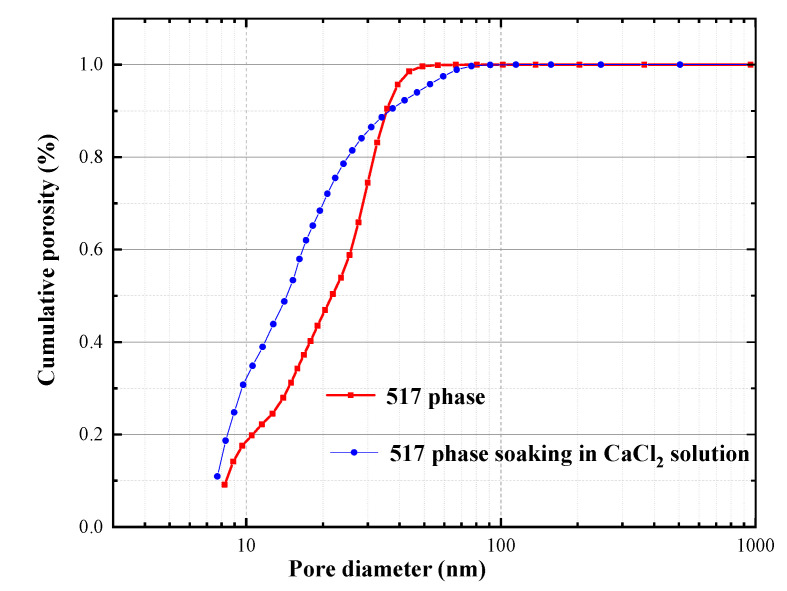
Cumulative porosity of hardened paste of 517 phase before and after immersion in CaCl_2_ solution.

**Figure 10 materials-13-05659-f010:**
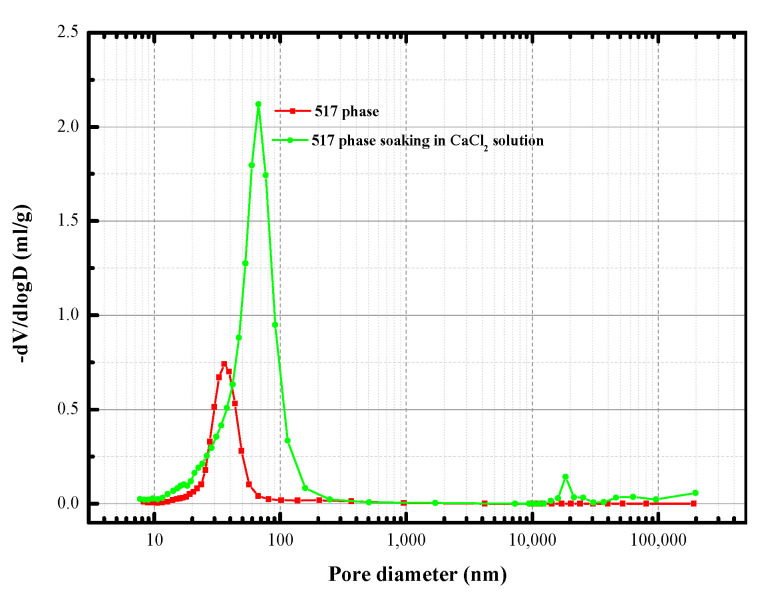
Pore size distribution of 517 phase before and after immersion in CaCl_2_ solution.

**Figure 11 materials-13-05659-f011:**
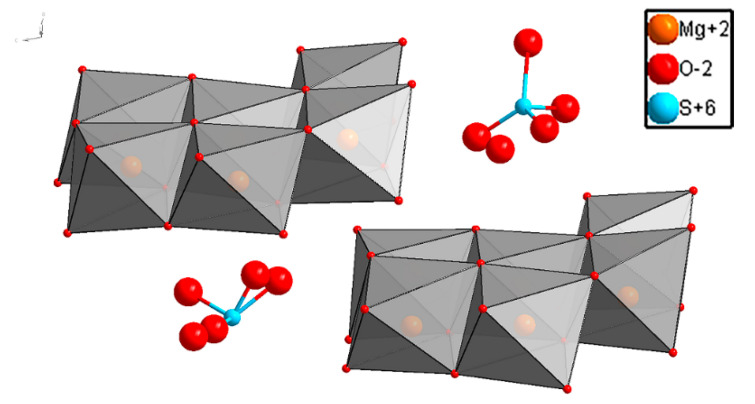
Crystal structure of 517 phase.

**Table 1 materials-13-05659-t001:** Contents of hardened paste of 517 phase after immersion in different chloride solutions for 28 days.

Chloride Species	Phase Composition (wt.%)			
	517 Phase	Mg(OH)_2_	CaSO_4_·2H_2_O	Amorphous Content
Before Soaking	92.84	0	/	7.16
NaCl	92.03	0	/	7.97
CaCl_2_	25.88	30.47	10.43	33.22
AlCl_3_	91.48	0	/	8.52
FeCl_2_	91.65	0	/	8.35
FeCl_3_	91.78	0	/	8.22

**Table 2 materials-13-05659-t002:** Concentrations of Mg^2+^ and SO_4_^2−^ in different chloride solutions used to immerse hardened paste for 28 days.

Chloride Species	NaCl	CaCl_2_	AlCl_3_	FeCl_2_	FeCl_3_
Mg^2+^ (mmol/L)	3.97	37.43	4.03	3.87	3.93
SO_4_^2−^ (mmol/L)	0.63	32.48	0.67	0.66	0.64

**Table 3 materials-13-05659-t003:** Standard molar thermodynamic properties of solid phases at 298.15 K.

Solids	$\Delta_{f}G_{m}^{\Theta }/(\mathrm{KJ}\cdot {\mathrm{mol}}^{-1})$	$\Delta_{f}H_{m}^{\Theta }/(\mathrm{KJ}\cdot {\mathrm{mol}}^{-1})$	$S_{m}^{\Theta }/(J\cdot {\mathrm{mol}}^{-1}\cdot K^{-1})$	Ref.
Mg(OH)_2_	−831.31 ± 0.11	−921.62 ± 0.11	63.18	[21]
517 Phase	−7026.43 ± 7.40	−7998.61 ± 3.60	674.32 ± 13.00	[21]
CaSO_4_·2H_2_O	−1797.62	−2032.44	162.55	[22]
Ca^2+^ (aq)	−553.60	−542.80	−53.10	[22]
H_2_O (l)	−237.13	−285.83	69.91	[22]
Mg^2+^	−454.80	−466.90	−138.10	[22]
SO_4_^2−^ (aq)	−744.50	−909.30	210.10	[22]

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
