# Peer review of "Effect of Ion Corrosion on 517 Phase Stability"

_materials, 2020, doi:10.3390/ma13245659_

Round 1
Reviewer 1 Report
The manuscript dealt with the ion corrosion on magnesium oxysulfate cement. Multiple characterization methods were adopted to study the properties and composition of the hardened pastes containing the 517 phase, and the influence of various types of chloride solutions on their stability. The study has confirmed that immersion of the 517 phase in CaCl2 solution could decompose the 517 phase and increase the porosity of the hardened pastes which is responsible for the observed reduction of the compressive strength. The manuscript was well organized and provided some useful data regarding the ion corrosion on the 517 phase. Some comments are provided for the authors’ consideration:
(1) At Page 7, Paragraph 1, Line 4-6, the authors described the increase of the concentration of magnesium and sulfate ions in the immersion solutions. This must be the case of CaCl2 solution. Please rephrase the sentence.
(2) The authors attributed the high concentration of magnesium and sulfate ions in CaCl2 solution as shown in Table 2 to the accelerated decomposition of the 517 phase. It is better that the authors could comment on the molar ratio of these two ions being 1 and how does ratio relate to the reaction presented in Eq. 4 and 5.
(3) In addition to discussion on the shift of the peak in pore size distribution presented in Figure 9, the authors may have also noticed that there is a second peak of pore size density distribution of 517 phase after immersion in CaCl2 solution. Does the second peak (~10 – 100 µm pores) relate to the presence of cracks? Please add a few sentences to comment on this.
(4) It is not clear what the authors were trying to say by presenting Fig. 10. Please clarify.
Reviewer 2 Report
This study investigated the effects of different types of chloride solution on the mechanical property and phase stability of the 517 phase in magnesium oxysulfate cement. Some interesting results are found in this study. The results showed that the stability of the 517 phase tends to be decomposed in CaCl2 solution, which leads to the decrease of the compressive strength of the 517 phase in MOS. The NaCl, AlCl3, FeCl2, FeCl3 solution has no significant effects on the compressive strength and phase stability of the MOS cement. There are only several minor issues that need to be addressed:
- The significance and objectives of this study should be specified in the manuscript.
- Since the results showed that the compressive strength of the 517 phase has no significant difference in samples immersed in NaCl, AlCl3, FeCl2, and FeCl3 solutions, the error bars should be added in Figure 8 to show the accuracy of the statements.
Reviewer 3 Report
The study is very interesting and with good quality. However some minor issues, need to be solved before its publication:
1) In 2.1 - preparation of raw materials is missing the concentration of each chloride solutions, and this is very important to understand the result achieved;
2) In the test methods is needed the standard/norms used for each test, and even when it is described as internal standard, needs to be better explained;
3) In the results section, a more detailed discussion is needed. It is not enough to present the results, but it is important discussed them. More important in the case of the CaCl2 solution results; or included this discussion on section 4.
4) The conclusions need to be improved.
Reviewer 4 Report
The manuscript reports a study focused on the influence of different types of chloride solutions on the stability and compressive strength of the 517 phase. The manuscript is within the scope of journal Materials. However, before publication some major concerns must be addressed, namely:
Abstract - Avoid the use of acronyms through the abstract that are not used after within the abstract.
Introduction - The acronyms used through the introduction must be defined in the introduction. The introduction must be deeply improved since in the present form it does not reflect the actual state of the art.
Section 2.1 - Please introduce a schematic of the preparation of raw materials and hardened paste.
Figure 3 - Please introduce the error bars.
The average pore size change before and after immersion should be deeply explored and explained.
Table 3 - Please use the same number of significant numbers
The compressive strength of the 517 phase decreased in the CaCl2 solution. This should be deeply explored and explained.
Round 2
Reviewer 4 Report
Most of the issues raised were well addressed. However, some minor concerns must be improved before publication:
Abstract - As far as I am concerned the use of acronyms that are not mentioned after being defined should be deleted since only make the abstract complicated and difficult to read.
Introduction - The improvements introduced are not enough. The introduction as presented does not reflect the actual state of the art.
